# Circulating miRNAs as Biomarkers of Obesity and Obesity-Associated Comorbidities in Children and Adolescents: A Systematic Review

**DOI:** 10.3390/nu11122890

**Published:** 2019-11-27

**Authors:** Maddi Oses, Javier Margareto Sanchez, Maria P. Portillo, Concepción María Aguilera, Idoia Labayen

**Affiliations:** 1Institute for Innovation & Sustainable Development in Food Chain, Public University of Navarre, Jeronimo de Ayanz Building, Campus de Arrosadia, 31006 Pamplona, Spain; idoia.labayen@unavarra.es; 2TECNALIA, Parque Tecnológico de Álava, Leonardo Da Vinci, 11, 01510 Miñano, Spain; javier.margareto@tecnalia.com; 3Nutrition and Obesity Group, Department of Nutrition and Food Science, Public University of the Basque Country (UPV/EHU) and Lucio Lascaray Research Institute, 01006 Vitoria, Spain; mariapuy.portillo@ehu.eus; 4CIBEROBN (Physiopathology of Obesity and Nutrition Network CB12/03/30038), Health Institute Carlos III (ISCIII), 28029 Madrid, Spain; caguiler@ugr.es; 5Instituto de Investigación Sanitaria BIOARABA, 01006 Vitoria, Spain; 6Department of Biochemistry and Molecular Biology II, Institute of Nutrition and Food Technology (INYTA), Biomedical Research Centre (CIBM), University of Granada, Avda. del Conocimiento s/n, 18016 Armilla, Spain; 7Instituto de Investigación Biosanitaria IBS.GRANADA, Complejo Hospitalario Universitario de Granada, 18014 Granada, Spain

**Keywords:** miRNAs, childhood obesity, biomarkers

## Abstract

Early detection of obesity and its associated comorbidities in children needs priority for the development of effective therapeutic intervention. Circulating miRNAs (microRNAs) have been proposed as biomarkers for obesity and its comorbidities; therefore, we conducted a systematic review to summarize results of studies that have quantified the profile of miRNAs in children and adolescents with obesity and/or associated disorders. Nine studies aiming to examine differences in miRNA expression levels between children with normal weight and obesity or between obese children with or without cardiometabolic diseases were included in this review. We identified four miRNAs overexpressed in obesity (miR-222, miR-142–3, miR-140-5p, and miR-143) and two miRNAs (miR-122 and miR-34a) overexpressed in children with obesity and nonalcoholic fatty liver disease (NAFLD) and/or insulin resistance. In conclusion, circulating miRNAs are promising diagnostic biomarkers of obesity-associated diseases such as NAFLD and type 2 diabetes already in childhood. However, more studies in children, using massive search technology and with larger sample sizes, are required to draw any firm conclusions.

## 1. Introduction

Childhood obesity is one of the most serious public health challenges of the 21st century [1]. In 2016, it was estimated that 41 million children under the age of five and over 340 million children and adolescents aged 5–19 had overweight or obesity [2]. Childhood obesity is associated with the development of type 2 diabetes mellitus (T2D), nonalcoholic fatty liver disease (NAFLD), metabolic syndrome (MetS), dyslipidemia, and cardiovascular diseases (CVD), later in life and already in childhood [3]. However, overweight and obesity, as well as their related diseases, are largely preventable; therefore, the prevention of childhood obesity and the early diagnosis of its associated diseases need to take high priority [1]. 

Biomarkers are measurable and quantifiable biological parameters which serve as indices for health and physiology-related assessments, such as disease risk and diagnosis, psychological disorders, metabolic processes and abnormalities, etc. [4]. Thus, biomarkers are useful to diagnose diseases or the susceptibility to suffer them. In this respect, the study and identification of biomarkers associated with obesity, T2D, and CVD may be useful for early identification, proper treatment, and good life assurance [5].

MicroRNAs (miRNA) are short, 21–23 nucleotides, single-stranded, noncoding RNA molecules that are encoded in the genomes of complex organisms [6]. MiRNAs are post-transcriptional gene-expression regulators that have been implicated in a wide variety of cellular processes and disease conditions [7,8]. Recently, miRNAs were established as biomarkers for several disease states [9] and have been repeatedly studied in the context of metabolic diseases [8,10,11].

Several systematic reviews have gathered information about miRNAs´ role in adipose tissue [12] and in the development of CVD [13] or T2D [14]. In recent years, a growing body of studies has determined the value of miRNAs as effective biomarkers to diagnose and assess the risk of obesity and its associated comorbidities. However, most studies have been focused on and conducted in adulthood; therefore, there is a lack of information regarding the role of miRNAs in childhood obesity. Considering the vast amount of information readily available on the regulatory roles of miRNAs, together with the current pandemic of pediatric obesity, miRNAs might be foreseen as useful biomarkers for the future development of effective strategies for early diagnosis and therapeutic intervention of pediatric obesity and its associated diseases [12,13,14]. Therefore, the purpose of the current systematic review is to hypothesize the biomarker role of circulating miRNAs in the early onset of obesity and associated comorbidities through the examination of available circulating miRNA profile data in children and adolescents with obesity, and in metabolic abnormalities related to obesity.

## 2. Materials and Methods

The systematic review was conducted by following the preferred reporting Items for Systematic Reviews and Meta-Analysis (PRISMA) statement and was registered in the International Prospective Register of Systematic Reviews (PROSPERO reference number CRD42019135051).

### 2.1. Search Strategy and Eligibility Criteria

We used the Population, Intervention, Comparison, Outcomes, and Study (PICOS) design tool to formulate the question and facilitate the literature search [15]. We conducted a literature search for all kinds of studies providing data on differences in miRNAs expression between children and adolescents with obesity and its related cardiometabolic diseases and controls. The studies were considered eligible for their inclusion if they met the following criteria: (1) the participants were between 3 and 19 years old; (2) they provided the quantified expression of miRNAs in the case and control groups; (3) the population of the studied group had overweight/obesity and/or associated metabolic diseases, such as T2D, insulin resistance (IR), NAFLD, and/or CVD risks factors; and (4) they were case and control studies or intervention studies. Studies that were not written in English or were gray literature, as well as reviews, editorials, opinions, letters, and meeting abstracts, were excluded.

### 2.2. Data Sources and Search Strategies

We conducted a systematic literature search in PubMed and the Web of Science database, selecting the original articles published until 23 November 2018. The keywords used in the search strategy were related to the following topics: (1) participants—children and adolescents; (2) comparison—miRNA expression; and (3) outcome—obesity and/or cardiometabolic diseases. Different search strategies were used for PubMed and Web of Science. Thereby, the search strategy for the PubMed database was as follows: (“children” OR “adolescent” OR “youth” OR “teenager” OR “boy” OR “girl” OR “kids” OR “preschoolers”) AND (“obesity” OR “adiposity” OR “metabolic risk” OR “cardiometabolic risk” OR “type 2 diabetes” OR “insulin resistance” OR “insulin sensitivity” OR “HOMA”) AND (“microRNA” OR “miRNA” OR “micro RNA” OR “microRNAs” OR “miRNAs” OR “circulating MicroRNA” OR “MiR”). The search strategy in Web of science database was: (“child*” OR “adolesc*” OR “youth” OR teen* OR boy* OR girl* OR kids*) AND (“obesity" OR “adiposity” OR “metabolic risk” OR “cardiometabolic risk” OR “type 2 diabetes" OR “insulin resistance" OR “insulin sensitivity" OR “HOMA”) AND (“microRNA” OR “miRNA” OR “circulating MicroRNA” OR “microRNAs” OR “miRNAs” OR “MiR”).

We found 102 scientific articles in the PubMed database and 172 in the Web of Science database. The 274 articles were imported into EndNote software (version X7, Thomson Reuters, PA, USA), and duplicate files were removed, at first, automatically by the software, and then by visual checking (see Figure 1).

### 2.3. Study Selection Process

Two independent reviewers (I.L. and M.O.) checked the 216 articles, after removing the duplicates. The titles and abstracts of these articles were examined to identify those that were likely to analyze the expression changes of miRNAs in children or young people with obesity, metabolic risk, CVD, T2D, or IR.

Those articles in which it was not possible to know their content by reading only the title or the abstract were read in full, in order to deliberate their final inclusion or exclusion in the systematic review. Disagreements about study selection were resolved by reaching consensus among reviewers. 

### 2.4. Data Collection Process and Data Items

One reviewer extracted the data from the included studies (M.O.), and the data’s accuracy was checked by a second reviewer (I.L.). A specific database was created in Excel (Microsoft Corp, Redmond, WA, USA).

The following fields were collected from each included study: (1) study (author identification and reference); (2) number of participants, age, and sex; (3) weight and cardiometabolic status of the two groups (cases and controls); (4) biological sample from where the miRNAs were extracted, miRNA search technique (global search or specific miRNA search), and the used laboratory technique to quantify miRNAs expression; and (5) differences on the miRNA expression, in each study, between cases and controls. Appendix A was reviewed in those cases where the full text did not provide all the relevant information needed for the data extraction.

### 2.5. Study Quality and Risk of Bias Assessment

Study quality was assessed by two independent reviewers (M.O. and I.L.) by using the systematical appraisal tool for cross-sectional studies, AXIS [16], that is recommended to address issues that are often apparent in cross-sectional studies [17]. AXIS assesses study quality by following different questions about introduction, methods, results, discussion, and others. Additionally, study quality was also examined by the “Quality Assessment Tool for Quantitative Studies” developed by the Effective Public Health Practice Project (EPHPP) [18].

## 3. Results

### 3.1. Study Selection and Characteristics

We identified seven studies (78%) examining differences in miRNA expression between children with normal weight and obesity [19,20,21,22,23,24,25], one study investigating differences in miRNA expression between children with obesity and with or without NAFLD [26], and one study comparing miRNAs expression between children with obesity, with or without IR [27].

The characteristics of the studies are shown in Table 1. Regarding age, two of the studies [19,20] included preschool participants, three of them [21,23,27] studied participants between 6 and 12 years of age, and the last four examined children and adolescents aged 6 to 18 years of age together [22,24,25,26]. The distributions of boys and girls in the cases and controls were similar in all the studies, except in the study of Prats-Puig et al. [23], whose participants were only boys. 

The participants were classified as cases or controls according to their BMI status, except in the study of Thompson et al. [26], in which they were classified according to their BMI and the presence or not of NAFLD, and in the study of Masotti et al. [27], in which all the participants had obesity and were assigned to the case or control group, depending on the presence or not of IR. 

In regard to the biological sample used, there were seven studies with circulating miRNAs obtained from plasma samples [19,21,22,23,25,26,27] and two studies that extracted miRNAs from peripheral blood mononuclear cells (PBMC) [20,24]. Finally, two different methods/approaches were used to profile miRNA expression: massive parallel sequencing (Illumina’s global miRNAs profiling workflow) or NanoString nCounter (microRNA panels) for global miRNA search, and TaqMan qPCR (quantitative polymerase chain reaction) for the study of a specific panel of miRNA set.

### 3.2. Risk of Bias Within Studies

The risk-of-bias assessment graph for the included studies is presented in Appendix A. Although some studies did not meet the overall objectives proposed in the introduction section, and some issues described in the methodology were not very clear, in general terms, all the studies showed an adequate systematic methodology to include in the article.

### 3.3. Results of Individual Studies

All the studies showed statistically significant differences in the expression level of specific miRNAs between cases and controls. Differences in miRNA expression levels were expressed/quantified in terms of fold change or the ratio of mean expression level in cases to mean expression level in controls for each miRNA studied. 

Seven studies [19,20,21,22,23,24,25] found significantly (*p* < 0.05) dysregulated miRNAs in the sample of children with obesity compared with their normal-weight peers (Appendix A). Three of them conducted a massive search of miRNAs [19,20,23]. Cui et al. [19], after selecting 18 miRNAs candidates by massive search, observed that eight of them were significantly dysregulated (*p* < 0.05) in plasma of children with obesity (Appendix A). Ouyang et al. [20] found eight miRNAs significantly (*p* < 0.05) dysregulated in PBMC of children with obesity (Appendix A). In the study of Prats-Puig et al. [23], the massive search selected 16 candidate miRNAs, and the authors found that 15 of them were significantly (*p* < 0.05) dysregulated in plasma of children with obesity (Appendix A). Only the miR-222 was consistently upregulated in two of these studies (see Table 2) [19,23].

Four studies conducted a specific search for miRNAs by qPCR sets [21,22,24,25]. Al-rawaf et al. (25) focused on a specific miRNA set of 10 miRNAs and observed that all of them were significantly (*p* < 0.05) dysregulated in plasma of children with obesity (Appendix A). Three of them (miR-222, miR-142-3p, and miR-140-5p) were also found to be overexpressed in the Prats-Puig et al. [23] study (Table 2), and miR-222 was found to be overexpressed in the Cui et al. study (Table 2) [19]. In contrast, miR-532-5p and miR-423-5p, which were found to be down-expressed in the study of Al-rawaf et al. [25], were up-expressed in the study of Prats-Puig et al. [23]. Similarly, the miR-146a that was found downregulated in the study of Al-rawaf et al. was reported as upregulated in the study of Cui et al. [19].

Iacomino et al. [21] followed a specific search strategy and selected a set of 372 miRNAs to be monitored. They found 8 miRNAs as significantly (*p* < 0.05) dysregulated in plasma of children with obesity (Appendix A). Can et al. [22], after a specific search for seven miRNAs, observed that six of them were significantly (*p* < 0.05) dysregulated in plasma of children with obesity (Appendix A). Carolan et al. [24] searched a specific set of three miRNAs and found that two of them were significantly (*p* < 0.05) dysregulated in PBMC of children with obesity.

The study of Thompson et al. [24] conducted a specific miRNA search for 20 miRNAs potentially involved in NAFLD and found 15 significantly (*p* < 0.05) dysregulated in the plasma of children with obesity and NAFLD compared with normal weight and non-NAFLD controls (Table 3). One of these miRNAs, mir191-5p, was also dysregulated in the study of Ouyang et al. [20]

In the study of Masotti et al. [27], the expression profile of plasma circulating miRNAs at fasting and after an oral glucose test tolerance (OGTT) was investigated (Appendix A). They conducted a specific miRNA search for 179 miRNAs in both situations. They found that 14 miRNAs were significantly (*p* < 0.05) dysregulated in fasting plasma of children with obesity and IR compared to children with obesity and insulin sensitivity (Table 4). Two of these miRNAs, miR-122-5p and miR-34a-5p, were also found dysregulated in the Thompson et al.´s study [26], and another one, miR-320a, was also reported dysregulated in Iacomino et al.´s study [21].

## 4. Discussion

In this systematic review, we aimed to identify and unify circulating miRNAs dysregulated in excess adiposity and obesity-associated metabolic abnormalities in children and adolescents. Our findings show the following: (i) there are still few studies focused on pediatric obesity and a low number of participants, and most of them use non-massive search methods for identifying dysregulated miRNAs; (ii) although there is a wide variability in the circulating miRNAs reported in the different studies, we can identify four circulating miRNAs, miR-222, miR-142-3, 140-5p, and miR-143 that are overexpressed in children with obesity; and that (iii) miR-122 and miR-34a seem to be overexpressed in children and adolescents with NAFLD and/or IR.

The analysis of previous data carried out in this review also unveils four miRNAs (miR-222, miR-142-3, 140-5p, and miR-143) as significantly overexpressed in children and adolescents with obesity in more than one report (Table 2). It is of note that miR-222, miR-142-3, and 140-5p were identified after a massive search [19,23] and that the results obtained in studies in adults are in agreement with these findings [28]. In this regard, elevated levels of these miRNAs were previously associated with higher BMI and were particularly upregulated in the presence of severe obesity [28]. In adults with morbid obesity, miR-142-3p, miR-140-5p, and miR-222 were related to adiposity markers, and, interestingly, their concentrations were substantially lowered after surgery-induced weight loss [28].

The identification of the miR-122 as potential biomarker of NAFLD in children with obesity is consistent with previous studies in adults and animal models [29]. The miR-122 is mostly expressed in the liver, and it regulates cholesterol production and hepatic function [30]. Indeed, in adults, miR-122 seems to be a key regulator of cholesterol and fatty acid metabolism in the liver [30], and it was associated with insulin resistance, obesity, metabolic syndrome, type 2 diabetes, and adverse lipid profile [31]. Moreover, in adults, high levels of circulating miR-122 have also been associated with increased concentrations of alanine aminotransferase (ALT), aspartate aminotransferase (AST), gamma-glutamyl transferase (GGT), triglycerides, and lower HDL-cholesterol levels, as well as with hepatic steatosis and the degree and progression of NAFLD [31,32,33,34]. In agreement with these results, Brandt et al. observed that miR-122 circulating (plasma or serum) levels were higher in children with NAFLD than in non-NAFLD overweight children and that miR-122 concentrations were associated with higher liver enzyme levels (i.e., ALT, AST, and GGT) [35]. Of note, in mice, the inhibition of miR-122 resulted in lower plasma cholesterol levels, halted hepatic lipid synthesis, and enhanced hepatic fatty acid oxidation [30]. Moreover, in adults, after bariatric surgery, hepatic function improvement significantly correlated with a decrease in circulating miR-122 levels [28]. These findings strengthen the role of miR-122 as a sensitive and specific blood biomarker of liver function. 

One study in young children [27] included in this review reported that the miR-122 was also associated with insulin resistance evaluated by means of an oral glucose tolerance test. In this line, some studies in adolescents, young adults, and older adults observed that circulating miR-122 levels were correlated with insulin resistance, suggesting that miR-122 could be involved in insulin resistance and might be used as a potential biomarker of diabetes risk [36,37,38] and progression [29]. Moreover, several genes targeted by the miR-122 have been implicated in the pathogenesis of IR, including genes involved in muscle responses to insulin, such as PRKAB1, a subunit of AMPK, that is a critical regulator of metabolism in IR [35,37,39]. However, it should be noted that the miR-122 is associated with high levels of triglycerides and cholesterol, and dyslipidemia is a common feature in patients with insulin resistance or diabetes [28,31,32,36]. Thus, Ye et al. observed that this miRNA was upregulated in patients who, in addition to type 2 diabetes, had NAFLD as compared with those who presented diabetes but not NAFLD [34].

In children and adolescents with obesity, the presence of NAFLD was associated with higher levels of miR-34a [26]. This finding agrees with previous reports in adults in which this miRNA was proposed as a useful diagnostic biomarker of NAFLD [40,41] and nonalcoholic steatohepatitis (NASH) in patients with NAFLD [41]. In addition, in animal models, hepatic miR-34a levels were elevated in dietary-obese mice and in ob/ob mice [42].

Previously reported data show that circulating miR-122 and miR-34a levels seem to be an extrahepatic biomarker of NAFLD and the progression of it, suggesting that both miRNAs might be able to serve as a noninvasive diagnostic marker against aggressive diagnostic methods, such as liver biopsy.

In conclusion, circulating miRNAs are promising diagnostic biomarkers of obesity-associated diseases, such as NAFLD and type 2 diabetes, already in childhood. However, it was not possible to identify a concrete miRNA profile in children with obesity. Likewise, the limited number of studies, the low number of participants, the lack of homogeneity in participants according to their stage of puberty, and the use of different techniques for the identification and quantification of miRNAs (specific extraction methods for example) may have influenced the high variability found in the miRNA profile reported by the included studies. Nevertheless, findings presented in the current review suggest that miR-122 and miR-34a may be overexpressed in children and adolescents with NAFLD and IR, and that circulating miR-222, miR-142-3, 140-5p, and miR-143 are overexpressed in children with obesity. However, more studies in children, using massive search technology and with larger sample sizes, are required to draw any firm conclusions. 

## Figures and Tables

**Figure 1 nutrients-11-02890-f001:**
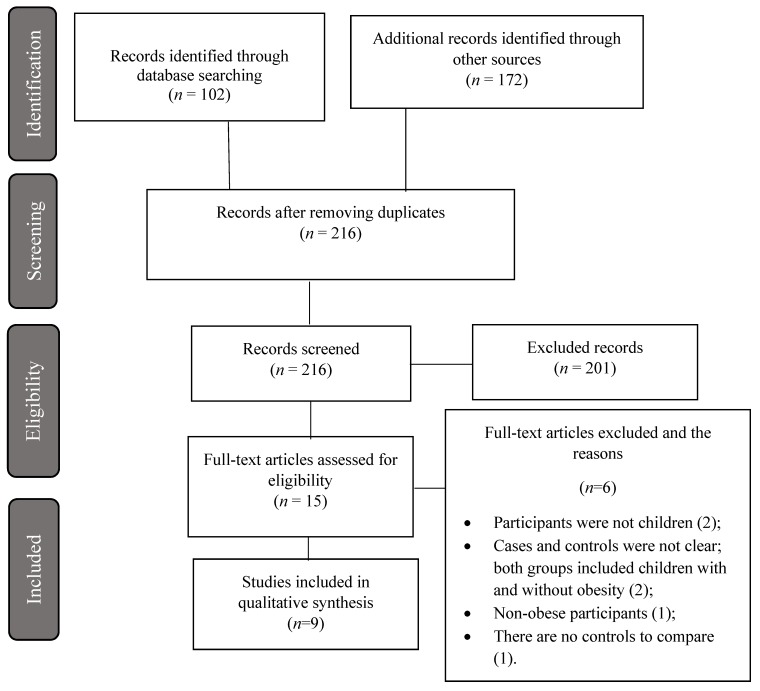
The PRISMA consort diagram for the search strategy. The initial search retrieved 274 articles, and a total of 9 studies were finally included after applying inclusion and exclusion criteria.

**Table 1 nutrients-11-02890-t001:** Characteristics of the studies examining differences in miRNA expression between obese and normal-weight children and between obese children with and without cardiometabolic risk factors.

AUTHOR	Study Aim	Biological Sample	Search for miRNA and Methodology		Participants	
			Type of Search Number of miRNAs Sample Size	Technique/Method	Cases Sample size Sex (girls %) Age BMI	Control Sample size Sex (girls %) Age BMI
Al-rawaf HA et al. 2018	To describe the circulating miRNA profile for adolescences and its association with circulating levels of leptin and adiponectin according to specific degree of obesity	Circulating miRNAs (Blood-plasma)	Specific search miRNAs *N* = 10 *N* = 150	qPCR	Obese 100 29 13.87 ± 2.91 years BMI = 26.7 ± 8.2 kg/m^2^	Normal weight 50 44 13.8 ± 2.88 years BMI = 17.4 ± 4.3 kg/m^2^
Cui et al. 2017	To screen candidate miRNAs as biomarkers for identifying obese children who are at risk of developing diabetes	Circulating miRNAs (Blood-plasma)	Massive - *N* = 18 Validation miRNAs *N* = 18 *N* = 246	Global miRNAs profiling - Illumina qPCR	Obese *N* = 100 51.5 61.0 ± 10.4 months BMI = 20.3 ± 2.2 kg/m^2^	Normal weight *N* = 146 49.5 60.4 ± 11.1 months BMI = 15.1 ± 1.06 kg/m^2^
Ouyang et al. 2017	To characterize the miRNA profile in PBMC of obese children	Circulating miRNAs (Blood—PBMC)	Massive - *N* = 12	Global miRNA Profiling - NanoString nCounter	Obese 50 39.7 ± 2.2 months BMI = 18.5 ± 26kg/m^2^	Normal weight 50 39.2 ± 2.3 months BMI = 13.5 ± 15 kg/m^2^
Thompson et al. 2017	To Evaluate whether circulating miRNAs that have been associated with NAFLD are altered in children with obesity, compared with healthy controls	Circulating miRNAs (Blood-plasma)	Specific search for miRNAs related to NAFLD miRNAs *N* = 20 *N =* 30	TaqMan RT-qPCR	Obese and NAFLD *N* = 20 42.8 13.2 ± 3.1 years BMI = 34.7 ± 10.4kg/m^2^	Normal weight and non-NAFLD *N* = 10 60 13.8 ± 2.1 years BMI = 20.1 ± 2.5 kg/m^2^
Iacomino et al. 2016	To identify circulating miRNAs potentially associated with early obesity in children	Circulating miRNAs (Blood-plasma)	Specific search miRNAs *N* = 372 *N* = 20	qPCR	Obese and overweight *N* = 10 40 10.7 ± 1.7 years BMI = 31,7 ± 4.3 kg/m^2^	Normal weight *N* = 10 50 10.5 ± 2.67 years BMI = 16.4 ± 1.7 kg/m^2^
Masotti et al. 2016	To investigate the expression profile of circulating miRNA 1) fasting and 2)120min after OGTT in 6 IR obese preschoolers and 6 controls without IR.	Circulating miRNAs (Blood-plasma)	Specific search miRNAs *N* = 179 *N* = 12	qPCR	Obese + IR *N* = 6 - 4.63 ± 1.82 years BMI = 20.9 ± 2.9 kg/m^2^	Obese without IR *N* = 6 - 4.35 ± 0.85 years BMI = 18.5 ± 1.2 kg/m^2^
Can et al. 2015	To examine the relationship between 7 specific miRNAs and lipid metabolism in obese and non-obese children and adolescents.	Circulating miRNAs (Blood-plasma)	Specific search miRNAs *N* = 7 *N* = 86	qPCR	Obese *N* = 45 57.7 14.71 ± 1.76 years BMI = 41.3 ± 52.9 kg/m^2^	Normal weight *N* = 41 58.5 14.44 ± 1.62 years BMI = 18.9 ± 2.1 kg/m^2^
Prats-Puig et al 2013.	To examine the dysregulated circulating miRNAs in obese children.	Circulating miRNAs (Blood-plasma)	Massive - *N* = 10 Validation miRNAs *N* = 15 *N* = 125	Global miRNA profiling – low-density TaqMan arrays (TLDAs) qPCR	Obese *N* = 5 0 (only boys) 8.8 ± 1.8 years z-BMI = 3.36 ± 0.43 Obese *N* = 40 55 9.2 ± 1.4 years z BMI = 2.69 ± 0.59	Normal weight N = 5 0 (only boys) 9.9 ± 1.0 years z-BMI = −0.62 ± 0.3 Lean *N* = 85 49 9.0 ± 1.6 years z BMI = −0.32 ± 0.71
Carolan et al. 2013	To investigate sCD163 levels, circulating iNKT frequency, cytokine profile and miR expression in obese and non-obese children.	Circulating miRNAs (Blood-PBMC)	Specific search miRNAs *N* = 3 *N* = 49	qPCR	Obese *N* = 29 46.4 13.0 ± 3.0 years z-BMI = 3.4 ± 0.5	Normal weight *N* = 20 35 12.8 ± 3.2 years z-BMI = 0.2 ± 1.1

PBMC: peripheral blood mononuclear cells, NAFLD: nonalcoholic fatty liver disease, OGTT: oral glucose tolerance test, IR: insulin resistance, qPCR: real-time polymerase chain reaction, TLDA: TaqMan low density arrays, BMI: body mass index, z-BMI: body mass index z-score.

**Table 2 nutrients-11-02890-t002:** Seven miRNAs expressions were altered in more than one study. Differences in miRNA expression between children with obesity and normal-weight children.

miRNAs	Author	Effect Size (Cases vs. Controls)			*p*
		Relative Expression Level	Mean Expression Level	Fold Change	
miR-222	Al-rawaf HA et al. 2018	14.5 vs. 4.5	-	-	<0.001
	Cui et al. 2017	-	-	>6	<0.01
	Prats-Puig et al. 2013	-	41.08 ± 30.59 vs. 25.43 ± 17.87	-	0.001
miR-142-3p	Al-rawaf HA et al. 2018	12 vs. 2.5	-	-	<0.001
	Prats-Puig et al. 2013	-	90.31 ± 61.46 vs. 32.30 ± 21.29	-	<0.0001
miR-140-5p	Al-rawaf HA et al. 2018	13.5 vs. 4	-	-	<0.001
	Prats-Puig et al. 2013	-	32.66 ± 18.13 vs. 23.15 ± 17.50	-	0.001
miR-143	Al-rawaf HA et al. 2018	14 vs. 3.5	-	-	<0.001
	Can et al. 2015	-	30.5 vs. 115.35	-	0.001
miR-532-5p	Al-rawaf HA et al. 2018	8 vs. 17	-	-	<0.001
	Prats-Puig et al. 2013	-	10.49 ± 7.75 vs. 5.49 ± 4.28	-	0.001
miR-423-5p	Al-rawaf HA et al. 2018	4 vs. 14	-	-	<0.001
	Prats-Puig et al. 2013	-	2.16 ± 1.35 vs. 1.13 ± 0.77	-	<0.0001
miR-146a	Al-rawaf HA et al. 2018	4 vs. 15	-	-	<0.001
	Cui et al. 2017	-	-	3.8	<0.01

**Table 3 nutrients-11-02890-t003:** Differences in miRNA expression between children with obesity and nonalcoholic fatty liver disease (NAFLD) and normal-weight and non-NAFLD children.

miRNAs	Author	Effect Size: Fold Change (Cases *vs* Controls)	*p*
miR-122-5p	Thompson et al. 2017	12.48	<0.0001
miR-34a-5p		5.09	<0.0001
miR-191-5p		7.21	<0.0001
miR-15b-5b		3.42	0.0004
miR-199a-5p		17.18	<0.0001
miR-222-3p		2.14	<0.0001
miR-223-3p		6.72	<0.0001
miR-181b-5p		3.29	0.0009
miR-23a-3p		5.3	<0.0001
miR-27b-3p		6.74	<0.0001
miR-21-5p		4.89	<0.0001
miR-451-5p		1.54	0.0404
miR-192-5p		3.78	<0.0001
miR-16-5p		1.56	0.0064
miR-29a-3p		2.81	<0.0001
miR-150-5p		1.79	0.0006
miR-214-5p		2.73	0.0213
miR-155-5p		2.63	0.0023
miR-103a-5p		3.38	<0.0001

**Table 4 nutrients-11-02890-t004:** Differences in miRNA expression between children with obesity and insulin resistance and children with obesity and insulin sensitivity.

Differences In miRNA Expression (fast).			
miRNAs	Author	Effect Size: Fold Change (Cases vs. Controls)	*p*
miR-122-5p	Masotti et al. 2016	2.82 ± 0.49	0.037
miR-34a-5p		2.41 ± 0.39	0.032
miR-320a		1.55 ± 0.11	0.014
miR-505-3p		3.11 ± 0,65	0.03
miR-26b-5b		1.63 ± 0.17	0.02
miR-146a-5p		1.48 ± 0.09	0.014
miR-148b-3p		1.47 ± 0.18	0.032
miR-342-3p		1.46 ± 0.25	0.05
miR-190a		−3.04 ± 0.39	0.032
miR-200c-3p		−2.78 ± 0.46	0.032
miR-205-5p		−2.60 ± 0.44	0.032
miR-95		−1.72 ± 0.26	0.032
miR-19a-3p		−1.55 ± 0.21	0.032
miR-660-5p		−1.50 ± 0.19	0.032

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
