# Peer review of "Circulating miRNAs as Biomarkers of Obesity and Obesity-Associated Comorbidities in Children and Adolescents: A Systematic Review"

_nutrients, 2019, doi:10.3390/nu11122890_

Round 1

Reviewer 1 Report

Review:

The review manuscript entitled “Circulating miRNAs as biomarkers of obesity and obesity and associated co-morbidities in children and adolescents: a systematic review” describes circulating miRNAs as promising diagnostic biomarkers of obesity associated diseases such as NAFLD and type 2 diabetes in childhood. There are very less systematic review describing role of miRNA as biomarker in pediatric obesity associated disorders, so there is a necessity of such review. In this review authors have adopted scientific search strategy presented with preferred reporting Items for Systematic Reviews and Meta-Analysis (PRISMA) consort diagram and quality of the studies are assessed through Appraisal systematically tool for Cross-Sectional studies (AXIS), providing strong evidence based conclusion. Although this review is a necessary addition for the audience to give a thorough recall of progress in this field, there are several editing mistakes in the review which should be taken care of to improve the consistency and to make the review more readable.

Line 27, 224, 228, 229, 281: 140-5p => miR-140-5p Line 38: q-PCR, real-time polymerase chain reaction => q-PCR, quantitative polymerase chain reaction Line 19, 67-78, 77, 97-100, 107, 238, 276, 277: correct font Line 140: presents => Presents Line 163: q-PCR (Real-Time Polymerase Chain Reaction => q-PCR Line 204: mir191-5p => miR-191-5p Table 3 and 4: heading of second column; Autor => Author

Suggestion: There may be more editing mistakes so it is advised that the review article should be checked by a native English speaker or journal’s editing services.

Response:

Minor Revision

Author Response

Dear Reviewer,

Please, find enclosed a revision of our manuscript, “Circulating miRNAs as biomarkers of obesity and obesity and associated co-morbidities in children and adolescents: a systematic review”. We would like to thank the Editorial committee for giving us the opportunity to revise and improve our manuscript, and the Editors and reviewers for their thoughtful and constructive comments.

We have considered all the suggestions and have incorporated them into the revised manuscript (two reviewers’ suggestions). Changes are highlighted in yellow in the last version of manuscript. We believe our manuscript is stronger as the result of these modifications. An itemized point-by-point response to the Editors and reviewers’ comments is presented below. 

The response is upload as a Word.

Thank you,

Reviewer 2 Report

This is a good topic for a systematic review but there are some inaccuracies in methodology which reduce the quality of the systematic review. Overall this manuscript has been submitted with errors in several places which show poor proof reading  

Title and Authors names: need some editing; poor proof reading

In several places different theme font has been used.

Methods:
Why was AXIS used for study quality and not a more appropriate tool suitable for RCTs and case control studies? This is major methodological flow.
Table 1 should be in results
In PRISMA chart clarify why Cases and controls were not clear
Table 2, 3 is difficult to read and requires formatting for further clarity

Author Response

(The authors gave the same response as above.)
